# Axion insulator state in hundred-nanometer-thick magnetic topological insulator sandwich heterostructures

Deyi Zhuo [1,4], Zi-Jie Yan [1,4], Zi-Ting Sun [2,4], Ling-Jie Zhou [1], Yi-Fan Zhao [1], Ruoxi Zhang [1], Ruobing Mei[1], Hemian Yi[1], Ke Wang[3], Moses H. W. Chan [1], Chao-Xing Liu [1], K. T. Law [2] ✉ & Cui-Zu Chang [1] ✉

An axion insulator is a three-dimensional (3D) topological insulator (TI), in which the bulk maintains the time-reversal symmetry or inversion symmetry but the surface states are gapped by surface magnetization. The axion insulator state has been observed in molecular beam epitaxy (MBE)-grown magnetically doped TI sandwiches and exfoliated intrinsic magnetic TI $MnBi_2Te_4$ flakes with an even number layer. All these samples have a thickness of ~10 nm, near the 2D-to-3D boundary. The coupling between the top and bottom surface states in thin samples may hinder the observation of quantized topological magnetoelectric response. Here, we employ MBE to synthesize magnetic TI sandwich heterostructures and find that the axion insulator state persists in a 3D sample with a thickness of ~106 nm. Our transport results show that the axion insulator state starts to emerge when the thickness of the middle undoped TI layer is greater than ~3 nm. The 3D hundred-nanometer-thick axion insulator provides a promising platform for the exploration of the topological magnetoelectric effect and other emergent magnetic topological states, such as the high-order TI phase.

An axion is a hypothetical particle postulated to resolve the charge conjugation-parity problem in particle physics[1,2]. It is also an attractive yet unobserved candidate for the missing dark matter in cosmology. Recent theoretical studies find the same mathematical structure of axion electrodynamics, specifically, a coupling term with the form $\theta e^2 \vec{E} \cdot \vec{B}/2\pi\hbar$, where $\theta$ plays the role of "axion", in a class of topological materials, called axion insulators[3–6]. Here $\vec{E}$ and $\vec{B}$ are electric and magnetic fields inside a material, respectively. Unlike the hypothetical axions in high-energy physics, the axion insulator has been realized in a three-dimensional (3D) topological insulator (TI), when its top and bottom surfaces have antiparallel magnetization alignment[3–5]. The Hall conductance in an axion insulator on the top and bottom surfaces cancels each other but the value of $\theta$ is pinned to $\pi$ by time-reversal symmetry and/or inversion symmetry in the interior. One

consequence of the quantized non-zero value of $\theta$ is the topological magnetoelectric (TME) effect, which refers to the quantized magnetoelectric response of the induced $\vec{E}$ to applied $\vec{B}$ and vice versa[3–8]. Besides the TME effect, certain types of axion insulators have been theoretically predicted to host a higher-order TI phase with chiral hinge states[9,10].

The axion insulator state has been observed in both molecular beam epitaxy (MBE)-grown magnetically doped TI sandwiches[11,12] and exfoliated intrinsic magnetic TI $MnBi_2Te_4$ flakes with an even number layer[13]. Although the axion insulator state by definition must be in a 3D regime[11,14], all samples showing the axion insulator signatures to date have a thickness limited to ~10 nm, near the 2D-to-3D boundary[3,15–18]. In transport measurements, the axion insulator usually shows a zero Hall conductance plateau under antiparallel magnetization alignment

[1]Department of Physics, The Pennsylvania State University, University Park, PA 16802, USA. [2]Department of Physics, Hong Kong University of Science and Technology, Clear Water Bay, 999077 Hong Kong, China. [3]Materials Research Institute, The Pennsylvania State University, University Park, PA 16802, USA. [4]These authors contributed equally: Deyi Zhuo, Zi-Jie Yan, Zi-Ting Sun. ✉e-mail: phlaw@ust.hk; cxc955@psu.edu

and a well-quantized quantum anomalous Hall (QAH)/Chern insulator state under parallel magnetization alignment[3,11–13]. Therefore, in the literature, the appearance of the zero Hall conductance plateau is frequently associated with the realization of the axion insulator state[3,11–13]. However, the zero Hall conductance plateau can also be formed by the hybridization gap between the top and bottom surfaces when the magnetic TI sandwich structure is thin[19,20]. Therefore, the appearance of only a zero Hall conductance plateau is not sufficient evidence for the axion insulator phase[3,15,16]. A recent study[16] suggests that the zero Hall conductance plateau observed in vanadium (V)-doped TI/TI/chromium (Cr)-doped TI sandwiches[11,12] is a result of the hybridization gap in the 2D regime rather than the formation of the axion insulator state in the 3D regime. Therefore, the observation of the axion insulator state in a thick sample clearly in the 3D regime is an important experimental priority and provides a favorable material platform for the exploration of the TME effect and the high-order TI phase.

In this work, we employ MBE to grow asymmetric magnetic TI sandwich heterostructures by systematically varying the thickness $m$ of the middle undoped TI layer, specifically, 3 quintuple layers (QL) $(Bi,Sb)_{1.89}V_{0.11}Te_3/m$ QL $(Bi,Sb)_2Te_3/3$ QL $(Bi,Sb)_{1.85}Cr_{0.15}Te_3$. Note that the thickness of 1 QL TI is ~1 nm. Our transport measurements confirm the axion insulator state with both the zero Hall resistance and conductance plateaus persists in the $m = 100$ sample with a total thickness of ~106 nm. The appearance of the well-quantized QAH effect in the $m = 100$ sample under high magnetic fields indicates that the side surfaces of the thick axion insulator are horizontally insulating and thus gapped. By varying the thickness $m$ of the middle undoped TI layer, we find that the axion insulator state starts to appear for $m \geq 3$. We also find that the two-terminal resistance in the axion insulator regime decreases rapidly with increasing $m$. We perform theoretical calculations on the side surface gap $\delta$ and find that its decay behavior with increasing $m$ is consistent with our experimental observation.

All 3 QL $(Bi,Sb)_{1.89}V_{0.11}Te_3/m$ QL $(Bi,Sb)_2Te_3/3$ QL $(Bi,Sb)_{1.85}Cr_{0.15}Te_3$ sandwich heterostructures are grown on ~0.5 mm thick heat-treated $SrTiO_3(111)$ substrates in a commercial MBE

chamber (Omicron Lab10) with a base pressure lower than ~$2 \times 10^{-10}$ mbar (Methods, Supplementary Fig. 1). The Bi/Sb ratio in each layer is optimized to tune the chemical potential of the sample near the charge neutral point[21–25]. The electrical transport measurements are carried out in a Physical Property Measurement System (Quantum Design DynaCool, 1.7 K, 9 T) and a dilution refrigerator (Oxford Instruments, 70 mK, 8 T) with the magnetic field applied perpendicular to the sample plane. The mechanically scratched six-terminal Hall bars are used for electrical transport measurements. More details about the MBE growth, sample characterizations, and transport measurements can be found in Methods.

## Results

We first focus on the magnetic TI sandwich heterostructure with $m = 100$. Figure 1a shows the cross-sectional scanning transmission electron microscopy (STEM) image of the $m = 100$ sample and the corresponding energy-dispersive X-ray spectroscopy (EDS) mappings of V and Cr near its top and bottom surface layers, respectively. Our cross-sectional STEM image shows the thicknesses of the top V-doped TI, the middle undoped TI, and the bottom Cr-doped TI layers are ~3 QL, ~100 QL, and ~3 QL, respectively. The total thickness of the $m = 100$ sample is ~106 nm, easily in the 3D TI regime with no hybridization gap between two surface states as the surface state penetration depth of TI is only around a few nanometers[15,16]. The EDS mappings of the V and Cr near its top and bottom surface layers show the top and bottom layers of the $m = 100$ sample are separately doped with V and Cr (Fig. 1a). The larger difference of the coercive fields and the weaker interlayer coupling between the top V-doped and bottom Cr-doped TI layers in thick magnetic TI sandwich will favor the formation of the axion insulator state, but the potential metallic side surfaces are expected to smear the appearance of the axion insulator state[3].

Next, we perform electrical magneto-transport measurements on the $m = 100$ sample at the charge neutral point $V_g = V_g^0$ and temperatures down to $T = 70$ mK (Fig. 1b, c). At $T = 70$ mK, the $m = 100$ sample shows the well-quantized QAH effect when the top and bottom

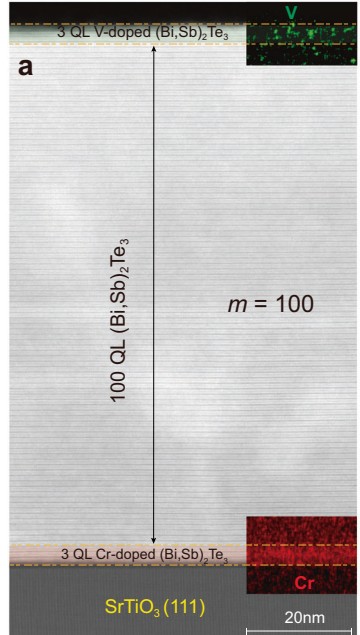
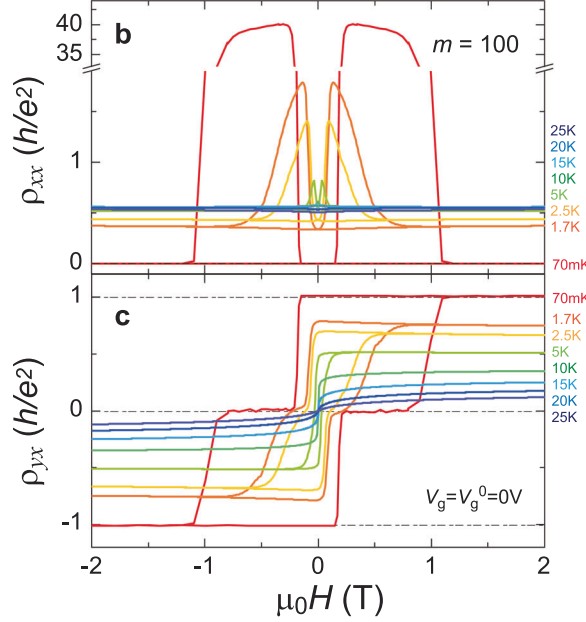

**Fig. 1 | MBE-grown 3 QL V-doped $(Bi,Sb)_2Te_3/100$ QL $(Bi,Sb)_2Te_3/3$ QL Cr-doped $(Bi,Sb)_2Te_3$ sandwich (i.e., the $m = 100$ sample). a** Cross-sectional STEM image. Inset: the EDS map of V (Cr) near the top (bottom) surface layers of the sample. **b, c** Magnetic field $\mu_0H$ dependence of the longitudinal resistance $\rho_{xx}$ (**b**) and the Hall resistance $\rho_{yx}$ (**c**) at $V_g = V_g^0 = 0$ V. At $T = 70$ mK and $V_g = V_g^0 = 0$ V, the observations of the zero $\rho_{yx}$ plateau and huge $\rho_{xx}$ between coercive fields of the top V- and the bottom Cr-doped TI layers indicate this $m = 100$ sample is in the axion insulator state.

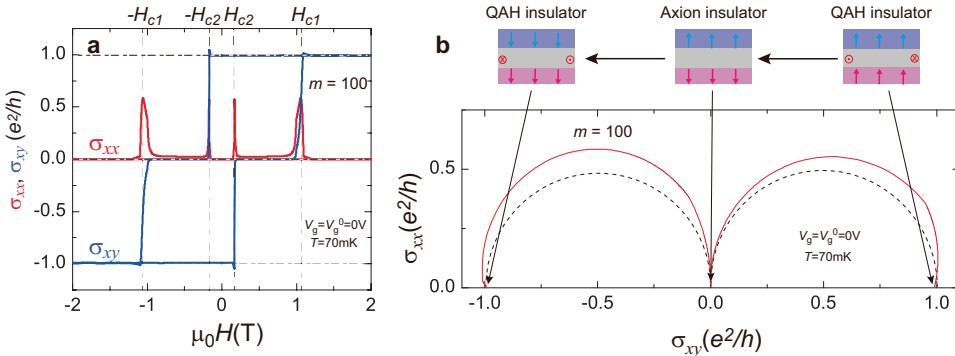

**Fig. 2 | Zero Hall conductance plateau and flow diagram of the $m = 100$ sample.**
**a** $\mu_0 H$ dependence of the longitudinal conductance $\sigma_{xx}$ (red) and the Hall conductance $\sigma_{xy}$ (blue) at $V_g = V_g^0 = 0$ V and $T = 70$ mK. **b** Flow diagram of $(\sigma_{xy}, \sigma_{xx})$ of the $m = 100$ sample. Two semicircles of radius $e^2/2h$ centered at $(e^2/2h, 0)$ and $(-e^2/2h, 0)$ are shown in dashed lines. Top: schematics of the QAH state at $(-h/e^2, 0)$ and $(h/e^2, 0)$ and the axion insulator state at $(0,0)$.

magnetically doped TI layers have the parallel magnetization alignment. Under zero magnetic field, the value of Hall resistance $\rho_{yx}$ is -1.008 $h/e^2$ (Fig. 1c), concomitant with longitudinal resistance $\rho_{xx} \sim 0.002\ h/e^2$ (-51 Ω) (Fig. 1b). The appearance of the well-quantized QAH state indicates that the side surfaces of the $m = 100$ sample are insulating along the sample edge direction and thus gapped, which is a prerequisite for the formation of the thick axion insulator state.

Besides the well-quantized QAH effect, the $m = 100$ sample also shows a clear zero Hall resistance $\rho_{yx}$ plateau between the coercive fields of the top V- and bottom Cr-doped TI layers. Here the coercive field $\mu_0 H_{c1}$ of the top V-doped TI layer is -1.058 T, while the coercive field $\mu_0 H_{c2}$ of the bottom Cr-doped TI layer is -0.171 T at $T = 70$ mK. Sweeping $\mu_0 H$ first reverses the magnetization of the bottom Cr-doped TI layer, resulting in the antiparallel magnetization alignment between the top V- and bottom Cr-doped TI layers. Therefore, the Hall resistance of the top and bottom surfaces of the $m = 100$ sample cancels each other and a zero Hall resistance $\rho_{yx}$ plateau appears between $\mu_0 H_{c1}$ and $\mu_0 H_{c2}$. The zero $\rho_{yx}$ plateau corresponds to the emergence of the axion insulator state, where a large value of $\rho_{xx}$ (>40 $h/e^2$) appears (Fig. 1b, c). Note that the axion insulator state can persist at $\mu_0 H = 0$ T through a minor loop measurement. With increasing $T$, the value of $\rho_{yx}$ gradually deviates from the quantized value, but the maximum value of $\rho_{xx}$ is greatly reduced and the zero $\rho_{yx}$ plateau is substantially shrunk. The maximum value of $\rho_{xx}$ is -1.787 $h/e^2$ and the value of $\rho_{yx}$ at $\mu_0 H = 0$ T is -0.785 $h/e^2$ at $T = 1.7$ K. The zero $\rho_{yx}$ plateau changes to a kink feature near the $\rho_{yx}$ sign change at $T = 5$ K (Fig. 1c). We note that the critical temperature of the QAH state in the $m = 100$ sample is ~4.5 K (Supplementary Fig. 2). Here the critical temperature of the QAH state is defined as that at which the ratio between the Hall and longitudinal resistances at $\mu_0 H = 0$ T is equal to 1 (Ref. 3). The two quantities track each other closely, indicating the intimate correlation between the zero $\rho_{yx}$ plateau under antiparallel magnetization alignment and the QAH effect under parallel magnetization alignment.

To further demonstrate the appearance of the axion insulator state in the $m = 100$ sample at $T = 70$ mK, we convert its $\rho_{yx}$ and $\rho_{xx}$ into Hall conductance $\sigma_{xy}$ and longitudinal conductance $\sigma_{xx}$. Four sharp peaks of $\sigma_{xx}$ are observed at $\pm\mu_0 H_{c1}$ and $\pm\mu_0 H_{c2}$ and two broad zero $\sigma_{xy}$ plateaus appear for $-\mu_0 H_{c1} \leq \mu_0 H \leq -\mu_0 H_{c2}$ and $\mu_0 H_{c2} \leq \mu_0 H \leq \mu_0 H_{c1}$ (Fig. 2a). The zero $\sigma_{xy}$ plateau and the corresponding nearly vanishing $\sigma_{xx}$ are a result of the cancellation of the contributions from the top and bottom surfaces in the antiparallel magnetization alignment, further confirming the realization of the axion insulator state in the $m = 100$ sample. To examine the magnetic field-induced quantum phase transition between QAH and axion insulator states, we plot the flow diagram $(\sigma_{xy}, \sigma_{xx})$ of the $m = 100$ sample at $T = 70$ mK (Fig. 2b). Two semicircles of radius $e^2/2h$ centered at $(\sigma_{xy}, \sigma_{xx}) = (\pm e^2/2h, 0)$ appear, the QAH and axion insulator states correspond to

$(\sigma_{xy}, \sigma_{xx}) = (\pm e^2/h, 0)$ and $(0, 0)$, respectively. For points $(\sigma_{xy}, \sigma_{xx}) = (-e^2/2h, 0.58\ e^2/h)$ and $(e^2/2h, 0.55\ e^2/h)$, which correspond to the quantum critical points with the appearance of extended states, the value of $\rho_{xx}$ is -0.98 $h/e^2$. This value is close to the universal value $h/e^2$, which has also been observed in the quantum phase transition between quantum Hall and a Hall insulator or QAH and an axion insulator[26,27]. This flow diagram validates the appearance of the zero $\sigma_{xy}$ plateau as a result of the cancellation of $\sigma_{xy} = \pm e^2/2h$ on top and bottom surfaces of the $m = 100$ sample.

To investigate the evolution of the zero $\sigma_{xy}$ plateau in magnetic TI sandwiches, we perform electrical transport measurements on 3 QL $(Bi,Sb)_{1.89}V_{0.11}Te_3/m$ QL $(Bi,Sb)_2Te_3/3$ QL $(Bi,Sb)_{1.85}Cr_{0.15}Te_3$ heterostructures by varying $m$ (Fig. 3, Supplementary Figs. 3 to 14). For $m \geq 12$, the sandwich samples show similar behaviors, i.e., the zero $\sigma_{xy}$ plateau and nearly vanishing $\sigma_{xx}$ for $\mu_0 H_{c2} \leq |\mu_0 H| \leq \mu_0 H_{c1}$ (Fig. 3a). The widths of the zero $\sigma_{xy}$ plateau are independent of the thickness of the middle undoped TI layer, which excludes the possibility of unintentional magnetic doping concentration difference. By further decreasing $m$, the strength of the interlayer magnetic coupling between the top V- and bottom Cr-doped TI layers becomes stronger, and the difference between $\mu_0 H_{c2}$ and $\mu_0 H_{c1}$ is reduced and thus the zero $\sigma_{xy}$ plateau becomes narrower and disappears for $m = 2$. The widths of the zero $\sigma_{xy}$ plateau are -0.70 T, -0.52 T, and -0 T for the $m = 8, 3$, and 2 samples (Fig. 3b–d). We find the zero $\sigma_{xy}$ plateaus in $m \geq 3$ samples persist at zero magnetic field by minor loop measurements (Supplementary Fig. 10). The disappearance of the zero $\sigma_{xy}$ plateau for the $m \leq 2$ samples indicates that the magnetizations of the top V- and bottom Cr-doped TI layers are strongly coupled, leading to the absence of the axion insulator state. For the $m = 1$ sample, the zero $\sigma_{xy}$ plateau completely disappears and the sample shows the standard QAH state[23,24,28–31]. Only one coercive field observed in the $m = 1$ sample indicates the collective flipping of the magnetizations of both top V- and bottom Cr-doped TI layers. We note that $\mu_0 H_c \sim 0.310$ T of the $m = 1$ sample is much larger than $\mu_0 H_{c2} \sim 0.171$ T of the $m = 100$ sample. This difference confirms the stronger interlayer exchange coupling with decreasing $m$ (Refs. 32–34). Remarkably, a unique feature here is that all samples over the very wide range of $m$, exhibit the well-quantized QAH state under high magnetic field, confirming that the side surface states are gapped along the sample edge direction.

Next, we plot the flow diagram $(\sigma_{xy}, \sigma_{xx})$ of the magnetic TI sandwich samples with different $m$ at $T = 70$ mK (Fig. 3g–l). The $m \geq 3$ samples show similar flow diagram $(\sigma_{xy}, \sigma_{xx})$ behavior as the $m = 100$ sample (Fig. 2b). Two semicircles of radius $e^2/2h$ centered at $(\sigma_{xy}, \sigma_{xx}) = (\pm e^2/2h, 0)$ appear, in which the QAH and axion insulator states correspond to $(\sigma_{xy}, \sigma_{xx}) = (\pm e^2/h, 0)$ and $(0, 0)$, respectively. With further decreasing $m$, the two semicircle behaviors gradually change to one semicircle of radius $e^2/h$ centered at $(0,0)$ for the

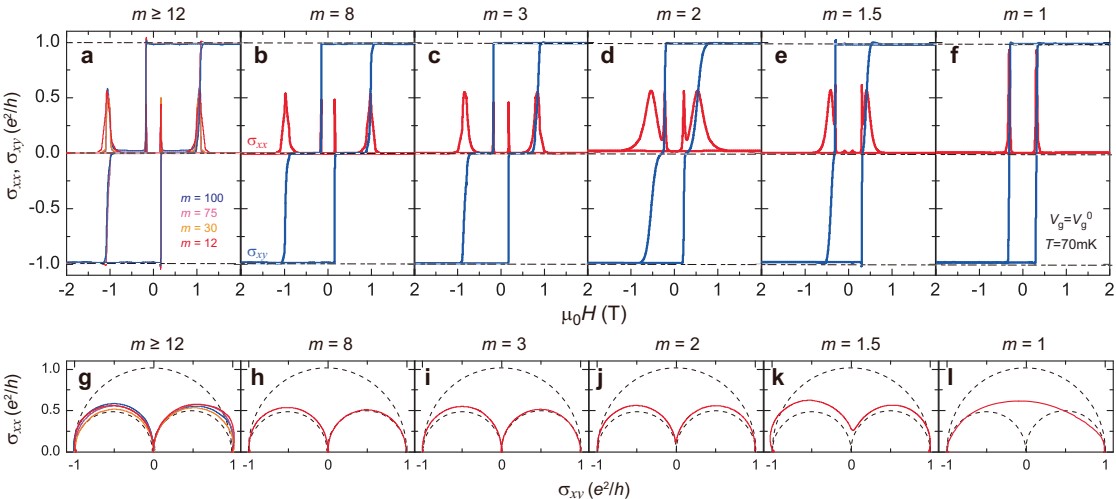

**Fig. 3 | Evolution of the zero Hall conductance plateau in magnetic TI sandwiches by varying *m*. a** $\mu_0 H$ dependence of $\sigma_{xx}$ and $\sigma_{xy}$ of the magnetic TI sandwiches with $m = 100$, $m = 75$, $m = 30$, and $m = 12$, respectively. **b–f** $\mu_0 H$ dependence of $\sigma_{xx}$ (red) and $\sigma_{xy}$ (blue) of the magnetic TI sandwiches with $m = 8$ (**b**), $m = 3$ (**c**), $m = 2$ (**d**), $m = 1.5$ (**e**), and $m = 1$ (**f**), respectively. **g** Flow diagrams of ($\sigma_{xy}$, $\sigma_{xx}$) of the

$m = 100$, $m = 75$, $m = 30$, and $m = 12$ samples. **h–l** Flow diagram of ($\sigma_{xy}$, $\sigma_{xx}$) of the $m = 8$ (**h**), $m = 3$ (**i**), $m = 2$ (**j**), $m = 1.5$ (**k**), and $m = 1$ (**l**) samples. Two semicircles of radius $e^2/2h$ centered at ($e^2/2h$, 0) and (-$e^2/2h$, 0) and one semicircle of radius $e^2/h$ centered at (0, 0) are shown in dashed lines. All measurements are taken at $V_g = V_g^0$ and $T = 70$ mK.

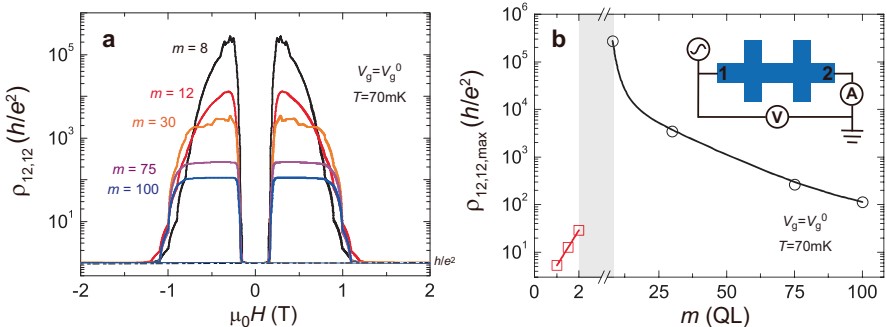

**Fig. 4 | *m* dependence of the two-terminal resistance of the axion insulator state in magnetic TI sandwiches. a** $\mu_0 H$ dependence of the two-terminal resistance $\rho_{12,12}$ of the axion insulator state in the $m \geq 8$ samples. **b** $m$ dependence of the maximum value of $\rho_{12,12}$ (labeled as $\rho_{12,12,max}$). The black circles and red squares indicate the $\rho_{12,12,max}$ values of the $m \geq 8$ and $m \leq 2$ samples, respectively. The

$\rho_{12,12,max}$ value exceeds the reliable range of our measurement setup in the gray shadow area. Inset: the measurements circuit we used to measure the huge $\rho_{12,12}$ of the axion insulator state. No series resistance is subtracted in these two-terminal measurements.

$m = 1$ sample. The disappearance of the (0, 0) point and the finite value of $\sigma_{xx}$ when $\sigma_{xy} = 0$ in the flow diagram for the $m \leq 2$ samples confirms the absence of the axion insulator state in these samples. The one semicircle flow diagram in our $m = 1$ sample with a total thickness of 7 QL also suggests that both the top V- and bottom Cr-doped TI layers are topologically nontrivial[21,22]. Therefore, the appearance of the zero $\sigma_{xy}$ plateau in our priors studies[11,26] should be a result of the antiparallel magnetization alignment rather than the formation of the hybridization gap in the $m = 4$, $m = 5$, and $m = 6$ samples.

We measure the thickness dependence of two-terminal resistance $\rho_{12,12}$ of our magnetic TI sandwiches with an alternating current (AC) voltage in the axion insulator regime (Fig. 4b inset)[26]. Figure 4a shows the $\mu_0 H$ dependence of $\rho_{12,12}$ of the magnetic TI sandwiches with $8 \leq m \leq 100$. In the QAH regime, the value of $\rho_{12,12}$ is -$h/e^2$ for all samples. However, in the axion insulator regime, there are two systematic trends as $m$ increases, which are absent in prior studies[11,12,26]. First, for the samples with $m \leq 30$, $\rho_{12,12}$ exhibits a pronounced peak as a function of $\mu_0 H$ in the axion insulator regime. However, for the $m = 75$ and $m = 100$ samples, $\rho_{12,12}$ changes slightly and shows a nearly flat feature as a function of $\mu_0 H$ in the axion insulator regime on a logarithmic scale

(Fig. 4a). The different $\mu_0 H$ dependence of $\rho_{12,12}$ in thin and thick axion insulators indicates that the response of the energy gap to magnetic fields varies in these two regions. As demonstrated with detailed calculations below, for thinner samples, the side surface states exhibit a large confinement gap and the energy gap is sensitive to the change of the magnetization gap of the surface states. On the other hand, for thicker samples, the energy gap is determined by the much smaller confinement gap of the side surfaces which is less sensitive to external magnetic fields. Second, all samples show much larger $\rho_{12,12}$ values (Fig. 4a) that increase with decreasing $m$. The maximum values of $\rho_{12,12}$ (labeled as $\rho_{12,12,max}$) are summarized in Fig. 4b. We find that the $\rho_{12,12,max}$ value of the $m = 8$ sample is -2.7 × $10^5$ $h/e^2$, which is three orders of magnitude larger than that of the $m = 100$ sample. For the $m = 3$ sample, the $\rho_{12,12,max}$ value exceeds the reliable range of our measurement setup. The $\rho_{12,12,max}$ value shows a sudden decrease for the $m \leq 2$ samples, specifically, $\rho_{12,12,max} \sim 29.2$ $h/e^2$, -12.5 $h/e^2$, and -5.3 $h/e^2$ for the $m = 2$, 1.5, and 1 samples (Fig. 4b and Supplementary Fig. 14), indicating the absence of the axion insulator state. To examine the QAH and the axion insulator phases in thick magnetic TI sandwiches, we measure the $V_g$ dependence of both the two-terminal

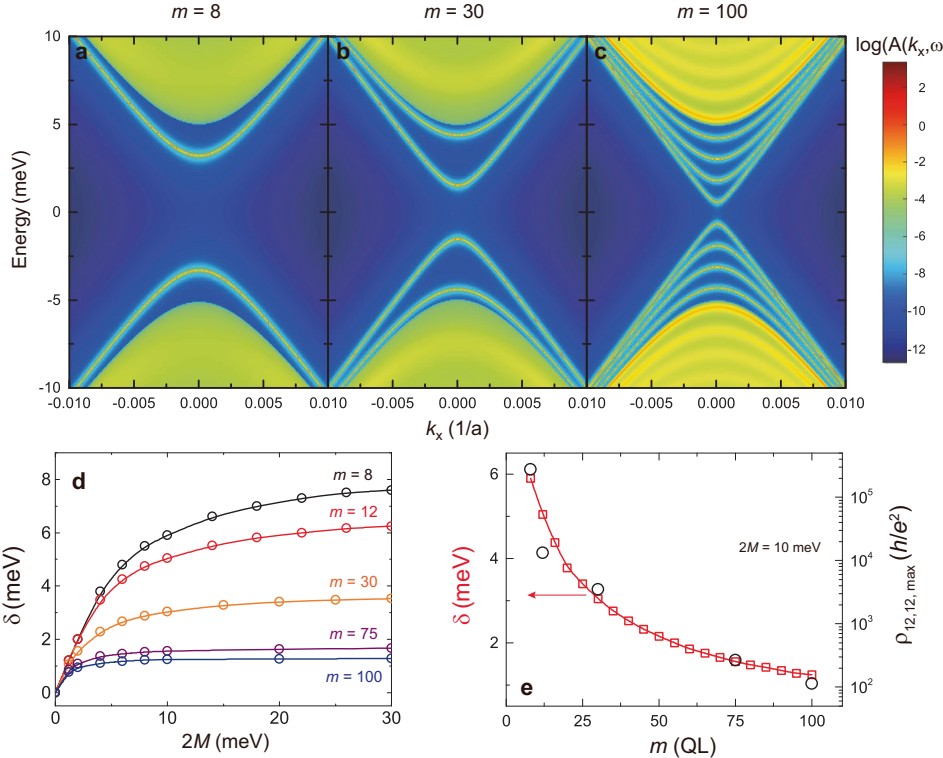

**Fig. 5 | Quantum confinement-induced surface gaps in thick axion insulators.** **a**–**c** Surface spectral functions in axion insulators with $m = 8$ (**a**), $m = 30$ (**b**), $m = 100$ (**c**). $2M = 10$ meV is used in **a**–**c**. **d** The side surface energy gap $\delta$ as a function of $2M$ in axion insulators with different $m$. **e** The side surface energy gap $\delta$ as a function of $m$ in axion insulators at $2M = 10$ meV. The black circles are the values of $\rho_{12,12,\text{max}}$ in Fig. 4b.

resistance and nonlocal resistance in the $m = 100$ sample when its top and bottom magnetic layers exhibit antiparallel and parallel magnetization alignments (Supplementary Figs. 11 and 12).

## Discussion

Our experiments on the $m = 100$ sample suggest that two key prerequisites for the axion insulator can be fulfilled: (i) the film thickness should be within the 3D regime, allowing the axion parameter to approach a quantized value and (ii) the side surface states should have a gap so that the material can exhibit an insulating behavior. For condition (i), a prior theoretical calculation[4] shows that the magnetoelectric parameter $\alpha$ that characterizes the axion term by $\alpha \frac{e^2}{2h} \vec{E} \cdot \vec{B}$ can exceed 0.9 when the thickness is greater than ~30 nm, and thus is expected to be even closer to the quantized value of 1 in our sample with a thickness of ~100 nm. This is an important reason why the observation of the axion insulating state in our thick samples is a crucial step for the investigation of the quantized magnetoelectric effect. For condition (ii), the rapid decrease of $\rho_{xx}$ with increasing temperature in the axion insulator regime (Fig. 1b) is typical behavior for an insulator phase, which implies a gap opening for the side surface states for the $m = 100$ sample. The origin of this side surface gap is likely attributed to the quantum confinement effect. We plot the spectral function of the side surfaces as a function of the momentum $k_x$ and energy $E$ for samples with different $m$ (Fig. 5a–c, and Supplementary Fig. 15). It is clear from Fig. 5a–c that there are dispersive side surface states, which are inherited from 3D TI. The confinement gap, partly caused by the finite thickness of the sample, becomes smaller as the thickness increases. Figure 5c, for $m = 100$, reveals a confinement gap of ~1.24 meV that resides within the magnetization gap of the top and bottom surface states which is about 10 meV. In the following, we study how the side surface gap changes as a function of the magnetization of the surfaces for different samples to explain the behaviors of $\rho_{12,12}$ in Fig. 4a in the axion insulator regime.

Figure 5d shows the side surface energy gap $\delta$ as a function of the magnetic exchange gap $2M$ for $m \geq 8$. We identify two typical regimes in these samples with varying $m$: (i) a linear region characterized by a significant slope of $\delta$ versus $2M$ at small $2M$ values; (ii) a saturated region where $\delta$ remains constant at large $2M$ values. These two regimes provide an understanding of the nearly flat feature observed in thick axion insulators and the peak behavior in thin axion insulators (Fig. 4a and Supplementary Fig. 13). At $2M = 10$ meV, the $m \leq 30$ samples fall within the linear region, indicating a dramatic change of $\delta$ when $2M$ varies, resulting in a peak feature of $\rho_{12,12}$ in the axion insulator regime. In contrast, the $m \geq 75$ samples are situated in the saturated region, where $\delta$ remains more robust against variations. Figure 5e shows the values of the side surface energy gap $\delta$ as a function of $m$ for a fixed $2M$. The calculated $\delta$ value reveals a decaying behavior with increasing $m$, similar to the thickness dependence of $\rho_{12,12,\text{max}}$ observed in our experiments (Fig. 4b) under a logarithmic scale. This similarity suggests that the side surface energy gap $\delta$ determines the excitation gap in this system. The qualitative and semi-quantitative agreement between our theoretical calculations and experimental results suggests that the confinement-induced gap is likely the primary source of the side surface gap. Note that the disorder effect may also play a role, but it will not alter the overall qualitative picture.

To summarize, we realize the axion insulator state showing the coexistence of the zero Hall resistance and conductance plateaus in a magnetic TI sandwich sample with a total thickness of ~106 nm. By varying the thickness of the middle undoped TI layer, we find that the 3 QL undoped TI layer is thick enough to produce a weak enough interlayer exchange coupling for the formation of the axion insulator state in magnetic TI sandwiches. The axion electrodynamics from the bulk $\theta$-term, which is unique in 3D, gives rise to many topological responses such as the topological magnetoelectric effect[4–6], image magnetic monopole[35], and quantized optical response[6]. Our

hundred-nanometer-thick magnetic TI sandwiches with the axion insulator state in the 3D regime (~106 nm thick) provide a better material platform for the exploration of these topological responses[4–6,35], as well as the higher-order TI phase[9,10]. Moreover, our thick magnetic TI sandwiches can also be employed to explore the existence of the half-quantized counter-propagating Hall current in axion insulators[36–39].

## Methods

### MBE growth

All magnetic TI sandwich heterostructures [i.e., 3 QL $(Bi,Sb)_{1.89}V_{0.11}Te_3$/$m$ QL $(Bi,Sb)_2Te_3$/3 QL $(Bi,Sb)_{1.85}Cr_{0.15}Te_3$] used in this work are grown in a commercial MBE system (Omicron Lab10) with a base vacuum better than ~$2 \times 10^{-10}$ mbar. To achieve satisfactory magnetization, both the top and bottom 3 QL $(Bi,Sb)_2Te_3$ layers doped with Cr or V are employed. The heat-treated insulating $SrTiO_3(111)$ substrates with a thickness of ~0.5 mm are first outgassed at ~600 °C for 1 hour before the growth of the magnetic TI sandwich heterostructures. High-purity Bi (99.9999%), Sb (99.9999%), Cr (99.999%), V (99.999%), and Te (99.9999%) are evaporated from Knudsen effusion cells to grow the samples. During the growth of doped and undoped TI, the substrate is maintained at ~230 °C and the Bi/Sb ratio is fixed at ~0.5 for all three layers. The flux ratio of Te per (Bi + Sb + Cr/V) is set to be greater than ~10 to prevent Te deficiency in the films. The Bi/Sb ratio in each layer is optimized to tune the chemical potential of the entire magnetic TI sandwich heterostructure near the charge neutral point. The growth rate of both magnetically doped TI and undoped TI films is ~0.2 QL per minute.

### Electrical transport measurements

All magnetic TI sandwich heterostructures grown on 2 mm × 10 mm insulating $SrTiO_3(111)$ substrates are scratched into a Hall bar geometry using a computer-controlled probe station. The effective area of the Hall bar is ~1 mm × 0.5 mm. The electrical ohmic contacts are made by pressing indium dots onto the films. The bottom gate is prepared by flattening the indium dots on the back side of the $SrTiO_3(111)$ substrates. Transport measurements are conducted using both a Physical Property Measurement System (Quantum Design DynaCool, 1.7 K, 9 T) for $T \geq 1.7$ K and a dilution refrigerator (Oxford Instruments, 70 mK, 8 T) for $T < 1.7$ K. The bottom gate voltage $V_g$ is applied using a Keithley 2450 meter. The excitation currents are 1 μA and 1 nA for the PPMS and the dilution four-terminal measurements, respectively. The two-terminal measurements are performed by a standard lock-in technique with a fixed voltage of ~0.1 mV. All magneto-transport results shown in this paper are symmetrized or anti-symmetrized as a function of the magnetic field to eliminate the influence of the electrode misalignment. The contact resistance including wires used in our dilution fridge at room temperature is ~34 Ω. No electronic filter is involved in our dilution refrigerator. More transport results can be found in Supplementary Figs. 2 to 14.

### Theoretical modeling

We perform theoretical calculations of the side surface gap based on a sandwich model[4]:

$$\mathcal{H}(\mathbf{k}) = \begin{bmatrix} M(\mathbf{k}) & -iA_1\partial_z & 0 & A_2k_- \\ -iA_1\partial_z & -M(\mathbf{k}) & A_2k_- & 0 \\ 0 & A_2k_+ & M(\mathbf{k}) & iA_1\partial_z \\ A_2k_+ & 0 & iA_1\partial_z & -M(\mathbf{k}) \end{bmatrix} + H_X \quad (1)$$

where $k_\pm = k_x \pm ik_y$, $M(\mathbf{k}) = M_0 + B_1\partial_z^2 - B_2(k_x^2 + k_y^2)$, and $H_X = \Delta(z)\sigma_z \otimes \tau_0$. To simulate the axion insulator state, we discretize it into a tight-binding model along the $z$-axis between neighboring QL from $\mathcal{H}(\mathbf{k})$. We also assume the spatial-dependent exchange field $\Delta(z)$ takes the values $M$ in the top three layers and $-M$ in the bottom three layers,

and zero in the middle $m$ layers, respectively. The parameters in our model are as follows: $M_0 = 0.28$eV, $A_1 = 2.2$eV · Å, $A_2 = 4.1$eV · Å, $B_1 = 10$eV · Å$^2$, $B_2 = 56.6$Å$^2$. The lattice constants are a = 4.14Å, c = 9.57Å. We choose the magnetic exchange gap $2M$ to be ~10 meV for our magnetic TI sandwiches. In our calculations, the side surface spectral function $A(k_x, \omega)$ is calculated from the recursive Green's function approach[40], which characterizes the density of states on the side surface (Fig. 5). We impose periodic boundary conditions along the $x$-direction with a good quantum number $k_x$, and calculate the semi-infinite lead surface Green's function $G_{1,1}$ along the $y$-direction to expose the side surface. By analyzing the side surface spectral function $A(k_x,\omega) = -\text{ImTr}(G_{1,1})$, we can determine the side surface energy gap δ.

## Data availability

The datasets generated during and/or analyzed during this study are available from the corresponding author upon request.

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

## Acknowledgements

We are grateful to Zhen Bi and Nitin Samarth for helpful discussions. This work is primarily supported by the NSF grant (DMR-2241327) (C.-Z.C. and C.-X.L.), including MBE growth, dilution transport measurements, and theoretical calculations. The sample characterization is supported by the ARO Award (W911NF2210159) (C.-Z.C.) and the Penn State MRSEC for Nanoscale Science (DMR-2011839) (C.-Z.C.). The PPMS measurements are supported by the DOE grant (DE-SC0023113) (C.-Z.C.). C.-Z.C. acknowledges the support from the Gordon and Betty Moore Foundation's EPiQS Initiative (GBMF9063 to C.-Z.C.). K.T.L. acknowledges the support from the Hong Kong Research Grants Council (RFS2021-6S03, C6025-19G, AoE/P-701/20, 16310520, 16310219, and 16307622).

## Author contributions

C.-Z.C. conceived and designed the experiment. Z.-J.Y. and Y.-F.Z. grew all magnetic TI sandwich samples and performed the PPMS measurements. D.Z. and L.-J.Z. fabricated the Hall bar devices and performed the dilution measurements. R.Z. and M.H. W.C. provided the technical support for the dilution measurements. K.W., H.Y. and Z.-J.Y. performed the STEM measurements. Z.-T.S., R.M., C.-X.L. and K.T.L. provided theoretical support. D.Z., Z.-T.S., C.-X.L., K.T.L. and C.-Z.C. analyzed the data and wrote the manuscript with input from all authors.

## Competing interests

The authors declare no competing interests.
