## [Peer Review File · Nature Communications]

REVIEWER COMMENTS

Reviewer #1 (Remarks to the Author):

This paper measures the transport signatures of the axion insulator states in thick magnetic topological insulator sandwich heterostructures. I agree with that author that the previous experimental evidences for the axion insulators remain elusive. They report that they employ MBE to synthesize magnetic TI sandwich heterostructures and find that the axion insulator state persists in a 3D sample with a thickness of ~ 106 nm, showing the coexistence of the zero Hall resistance and conductance plateaus in a magnetic TI sandwich sample. In my opinion, this is an important and impressive work which has significant impact on the field of the axion insulator. This paper is worth publishing in nature communication. However, I have the following concerns:

(1) In this work, the authors focus on the case when the Fermi energy lies inside the side surface gap (which is much smaller compared to that of the magnetic gap on the surface layers, especially for thick films). Therefore, in the axion insulator regime, the system manifests the typical behavior of an insulating phase, because there is no state. And in the "3D" QAH regime, the system manifests the typical behavior of a 2D QAH phase. With the increasing film thickness, there appears no significant difference as long as the Fermi energy is located in the side surface gap.

However, it is important to notice that the QAH phase and the axion insulator phase are manifested when the Fermi energy resides inside the magnetic gap, not the side surface gap. I think it is much more important to study the case when the Fermi energy crosses the side surface bands, considering a recent work [PRB 105, L201106 (2022)] has pointed out that it is the side surface states that are responsible for the chiral conducting currents of the axion insulator phases. On the other hand, for the QAH phase, when the Fermi energy crosses the side surface bands, the Hall resistance may deviate from the quantization, however, the Hall conductance should maintain the quantization [see Nature Physics, 18, 390 (2022)].

(2) I do not see a clear transport signature that could distinguish the axion insulator state in thick films ($m > 10$) and thin films ($10 > m > 3$), because the zero Hall conductance plateau cannot describe the coupling strength between the top and bottom surface layers. Whether such signatures may be observed by applying the nonlocal transport measurement [see Science Bulletin, 68, 1252 (2023)], based on existing equipment? If the half-quantized counter-propagating Hall current is successfully observed, the author's findings undoubtedly represent a significant advancement in the field of axion insulators.

(3) The author uses Hall/longitudinal conductance/resistance to characterize the system. However, in my opinion, the conductance/resistance varies when changing the system size. It may be better to use the conductivity/resistivity to characterize the system. This aspect may be much more complex for a 3D system, with the chiral currents located on either the 1D hinge or the 2D side surface.

(4) The author said, “We also find that the two terminal resistance in the axion insulator regime decreases nearly exponentially with increasing m , implying the rapid reduction of the transport gap, which presumably originates from the side surface states in thick samples”. Why the “nearly exponentially” decay implies a reduction of the transport gap?

(5) I suppose that the side surface gap should exhibit a power-law decay with the increase in the film thickness, which can be analytically confirmed by the quantum well approximation.

Reviewer #2 (Remarks to the Author):

This manuscript studies axion insulator state in magnetically doped topological insulator (TI) sandwiches of V-doped $(\text{Bi,Sb})_2\text{Te}_3$ / $(\text{Bi,Sb})_2\text{Te}_3$ /Cr-doped $(\text{Bi,Sb})_2\text{Te}_3$ with varying the middle undoped TI layer thickness up to 100 nm. The zero Hall conductance plateau is now regarded as insufficient evidence of the axion insulator state because identical features can be seen by other reasons such as the existence of the hybridization gap in the 2D regime. The observation, in this manuscript, of the zero Hall conductance plateaus from the thick samples in the 3D regime provide further evidence of the axion insulator states in the magnetically doped TI sandwiches. The results would be useful to the community. However, I would suggest publication of this manuscript in the Nature Communications after the authors address my comments and questions:

One of the main concerns in the 3D regime is conduction through the bulk and side surfaces, which need to be insulating to observe well-quantized quantum anomalous hall (QAH) “insulator” and axion “insulator” states. The authors stated that all the observed data are at the charge neutral point by tuning the back gate voltage. Also, they claim that the side surface states are gapped and insulating, and the energy gap opening for the side surface states that they discuss in Fig. 5 is less than 10 meV (1.24 meV for 100nm sample). Since the gap is small, this may need to be better supported by experimental evidence.

1. 100nm $(\text{Bi,Sb})_2\text{Te}_3$ is not quite thin for back gating. Gating near the bottom layers is more efficient than gating near the top layers. At the charge neutral point, would the chemical potential

near the bottom layers and near the top layers be the same for thick samples? Or more p-type near the bottom layers, and more n-type near the top layers, for example? Showing gate-dependent ρ_{xx} and ρ_{yx} will be helpful for the authors' statement regarding the bulk conduction.

2. It is stated that the energy gap for the side surface states is smaller in thicker samples ($\sim 1\text{meV}$ for $m=100$ sample and $\sim 6\text{meV}$ for $m=8$ sample). And, well-quantized QAH and axion insulator states are expected when the chemical potential is located within the gap. Narrower gate-voltage range for well-defined quantization due to smaller gap is anticipated in thicker samples. Can this be provided?

Below are other comments and questions:

3. In page 8 line 156, it is stated that "the side surface states are insulating". The wording of "surface states" and "insulating" sounds contradictory.

4. In page 8 line 167-169, "Therefore, the appearance of the zero σ_{xy} plateau in our priors studies should be a result of the antiparallel magnetization alignment rather than the formation of the hybridization gap in the $m=4$, $m=5$, and $m=6$ samples." - Do all the samples in the previous studies have the sample details and growth conditions such as doping concentrations, gate dependence, and so on? Otherwise, results in this manuscript cannot be generalized for all the previous studies by the authors.

5. From the two-terminal resistance measurements, it is stated that $\rho_{12,12}$ in the QAH regime is $\sim h/e^2$ for all samples. What is the contact resistance? Were there any additional series resistance, for example from the electronic filters in the cryostats? Did the authors subtract any series resistance?

6. In page 9 line 176, "near-plateau feature" – It looks flat in logarithmic scale, but it does not look like a plateau in linear scale.

7. In page 11 line 224, "the confinement-induced gap should be the main origin of the side surface gap" – This statement is too strong since it just came from the qualitative similarity of the decaying δ in linear scale with the decaying resistivity in logarithmic scale.

8. EDS mapping of V (green) and Cr (red) near V-doped and Cr-doped regions is shown in Fig. 1a. The colors in the doped layers and supposedly undoped layers look similar. Does this mean that the dopants diffused into the undoped layers?

9. In the supplementary Figs. 4-7 for $m=3$ to 30, some of the ρ_{yx} curves look very noisy and they go out of the given y-axis. How much do they deviate? What is the interpretation of the huge noise in the samples while some are not as noisy? Can this be replotted and seen in the full y-axis range instead of cropping them within $\pm 1 h/e^2$?

Reviewer #3 (Remarks to the Author):

This manuscript presents the demonstration of an axion insulator state in a 3D topological insulator heterostructure and the investigation of its properties as a function of thickness. This investigation is made possible by the MBE growth of a heterostructure with an insulating layer of variable thickness separating top and bottom layers doped with two different magnetic elements, permitting the realization of antiparallel magnetization orientations. The existence of the axion insulator state is verified by the presence of the quantum anomalous Hall effect along with a zero Hall conductance plateau near field. Varying the intermediate insulating layer thickness shows that the conditions for the axion insulator state disappear in the 2D limit of 1 QL. The authors also claim that the presence of the axion insulator state is dependent upon there being a gap in the intermediate layer side surface states.

The novelty of this work lies in the demonstration of the axion insulator state in thick 3D materials well away from the 2D limit (i.e. $m \gg 10$). Overall, the study thoroughly investigates the effect of the thickness of the intermediate layer on the transport behavior, and the manuscript presents a clear discussion of the origin of the phenomena that are experimentally observed. Furthermore, theory is used to support the assertion that the intermediate layer side surface states are gapped. While the manuscript points out that this study is unique in that it investigates the 3D axion insulator state in samples with thicknesses greater than 10 nm, The study itself indicates that this state is present in samples with thicknesses greater than 3 nm. To strengthen the novelty of this study, a discussion on the new information that is gained over those previous studies should explicitly be included.

The methodology appears to be sound, although more information should be included about the heterostructure to enable replication. The concentrations of V and Cr doping should be included, and the rationale for selecting 3 QL thicknesses for each of these doped layers should be noted.

A few additional minor questions should be addressed for improved comprehension:

- The experimental and theoretical investigation included samples with m up to 100. However, the theoretical calculations suggest that the gap in the side surface states (a condition for this effect) starts to close around $m = 100$. At what thickness does the 3D axion insulator state expected to cease to be observed?
- “ $2M$ ” should be defined in the text.

-----Response to reviewers' comments-----

Reviewer #1

This paper measures the transport signatures of the axion insulator states in thick magnetic topological insulator sandwich heterostructures. I agree with that author that the previous experimental evidences for the axion insulators remain elusive. They report that they employ MBE to synthesize magnetic TI sandwich heterostructures and find that the axion insulator state persists in a 3D sample with a thickness of ~ 106 nm, showing the coexistence of the zero Hall resistance and conductance plateaus in a magnetic TI sandwich sample. In my opinion, this is an important and impressive work which has significant impact on the field of the axion insulator. This paper is worth publishing in nature communication. However, I have the following concerns:

We thank Reviewer #1 for his/her concise summary and positive assessment of our work.

Comment 1:

(1) In this work, the authors focus on the case when the Fermi energy lies inside the side surface gap (which is much smaller compared to that of the magnetic gap on the surface layers, especially for thick films). Therefore, in the axion insulator regime, the system manifests the typical behavior of an insulating phase, because there is no state. And in the "3D" QAH regime, the system manifests the typical behavior of a 2D QAH phase. With the increasing film thickness, there appears no significant difference as long as the Fermi energy is located in the side surface gap.

However, it is important to notice that the QAH phase and the axion insulator phase are manifested when the Fermi energy resides inside the magnetic gap, not the side surface gap. I think it is much more important to study the case when the Fermi energy crosses the side surface bands, considering a recent work [PRB 105, L201106 (2022)] has pointed out that it is the side surface states that are responsible for the chiral conducting currents of the axion insulator phases. On the other hand, for the QAH phase, when the Fermi energy crosses the side surface bands, the Hall resistance may deviate from the quantization, however, the Hall conductance should maintain the quantization [see Nature Physics, 18, 390 (2022)].

Response: We agree with Reviewer #1 that a systematic investigation of the QAH and the axion insulator phases as a function of the chemical potential (i.e., gate voltage V_g) is an interesting and important project. We did measure V_g -dependent two-terminal resistance $\rho_{16,16}$ of the $m = 100$ sample when its top and bottom magnetic layers have antiparallel and parallel magnetization

alignments (Fig. R1). For the antiparallel magnetization alignment, $\rho_{16,16}$ shows a prominent peak at $V_g=0V$ and decreases dramatically when the chemical potential starts to cross the side surface bands (Fig. 1b). However, for the parallel magnetization alignment, the value of $\rho_{16,16}$ is $\sim h/e^2$ near $V_g = V_g^0$ and does not change much when the chemical potential crosses the helical side surface bands [Chang et al. *Phys. Rev. Lett.* **115**, 057206 (2015)]. In our experiments, we have no evidence to show that the side surface states are responsible for the chiral conducting current in the axion insulator phase [Zou et al. *Phys. Rev. B* **105**, L201106 (2022)] and the half-quantized Hall conductance σ_{xy} persists when its chemical potential crosses the side surface bands [Mogi et al. *Nat. Phys.* **18**, 390 (2022)]. More careful measurements are required to substantiate these two assertions in the future.

Fig. R1| Gate dependence of the two-terminal resistance in the $m=100$ sample with antiparallel and parallel magnetization alignments. a, Schematics of the magnetic TI sandwich Hall bar device. The current flows from 1 to 6. **b, c**, V_g dependence of $\rho_{16,16}$ under antiparallel (**b**) and parallel (**c**) magnetization alignments. All measurements are taken at $T=70$ mK and $\mu_0H=0T$.

Based on topological field theory, the topological response of a 2D QAH phase can be fully captured by a bulk Chern-Simons action $S_{\text{eff}} = \frac{C_1}{4\pi} \int d^2x \int dt A_\mu \epsilon^{\mu\nu\tau} \partial_\nu A_\tau$ with bulk Chern number C_1 , while the “3D” QAH phase in the current study is described by the so-called θ -term, $S_{3D} = \frac{1}{4\pi} \int d^3x \int dt \epsilon^{\mu\nu\sigma\tau} \theta(\mathbf{x}, t) \partial_\mu A_\nu \partial_\sigma A_\tau$, with axion field $\theta(\mathbf{x}, t)$ [Qi et al. *Phys. Rev. B* **81**, 159901 (2010)]. In the bulk of our thick QAH samples, $\theta(\mathbf{x}, t)$ is uniform and equals to π , while in the vacuum outside the system $\theta(\mathbf{x}, t)$ equals to 0. Nevertheless, near the top and bottom surfaces, because of the spatial gradient of $\theta(\mathbf{x}, t)$, S_{3D} is reduced to the surface Chern-Simons

action $S_{\text{surf}} = \frac{1}{4\pi} \int_{\partial V} d\hat{n}_\mu \left(g[\mathbf{M}(\vec{x})] + \frac{1}{2} \right) \epsilon^{\mu\nu\sigma\tau} A_\nu \partial_\sigma A_\tau$, where $g[\mathbf{M}(\vec{x})]$ takes 0 if the surface magnetization $\mathbf{M}(\vec{x})$ points outward at the surface and takes 1 if $\mathbf{M}(\vec{x})$ points inward at the surface. The combination of S_{surf} at the top and bottom surfaces is responsible for both the quantized σ_{xy} plateau of the QAH phase and the zero σ_{xy} plateau of the axion phase in our system. However, it is the bulk θ -term with $\theta = \pi$ that makes a difference. The axion electrodynamics from the bulk θ -term, which is unique in 3D, gives rise to many topological responses such as the topological magnetoelectric effect, image magnetic monopole, and quantized optical response [Qi et al. *Phys. Rev. B* **81**, 159901 (2010); Qi et al. *Science* **323**, 1184(2009); Nenno et al. *Nat. Rev. Phys.* **2**, 682(2020); Chang, Liu, and MacDonald, *Rev. Mod. Phys.* **95**, 011002 (2023)]. Our system exhibits well-quantized QAH and axion insulator behaviors (all surface states gapped) in the 3D regime (~ 106 nm thick) and thus provides an ideal platform for the exploration of these topological responses.

We added Fig. R1 and relevant discussion in the revised Supplementary Information.

Comment 2:

*(2) I do not see a clear transport signature that could distinguish the axion insulator state in thick films ($m > 10$) and thin films ($10 > m > 3$), because the zero Hall conductance plateau cannot describe the coupling strength between the top and bottom surface layers. Whether such signatures may be observed by applying the nonlocal transport measurement [see *Science Bulletin*, 68, 1252 (2023)], based on existing equipment? If the half-quantized counter-propagating Hall current is successfully observed, the author's findings undoubtedly represent a significant advancement in the field of axion insulators.*

Response: We respectfully disagree with Reviewer #1's statement "*the zero Hall conductance plateau cannot describe the coupling strength between the top and bottom surface layers.*". We believe that the width of zero σ_{xy} plateau can be used to estimate the coupling strength between the top and bottom surface magnetic TI layers in the current work. We first would like to clarify some confusion about the origin of the observed zero σ_{xy} plateau in magnetic TI films/heterostructures. For the zero σ_{xy} plateau observed in thin magnetic TI films/sandwiches [Kou et al. *Nat. Commun.* **6**, 8474 (2015); Feng et al. *Phys. Rev. Lett.* **115**, 126801 (2015)], Reviewer #1's statement is correct because the formation of the zero σ_{xy} plateau is an artifact due

to the conversion of the measured large longitudinal resistance (ρ_{xx}) and Hall resistance (ρ_{yx}) into

$$\sigma_{xy}, \text{ i. e. } \sigma_{xy} = \frac{\rho_{yx}}{\rho_{xx}^2 + \rho_{yx}^2} \text{ [Chang, Liu, and MacDonald, } Rev. Mod. Phys. \text{ } \mathbf{95}, 011002 \text{ (2023)]. We}$$

noted that for these thin films/sandwiches, the coupling strength between the top and bottom surfaces can also lead to the zero σ_{xy} plateau, which is difficult to distinguish from the anti-ferromagnetic alignment of magnetization discussed below. However, for the zero σ_{xy} plateau observed in thick magnetic TI sandwiches in the current work, the formation of the zero σ_{xy} plateau is a result of antiparallel magnetization alignment [Xiao et al. *Phys. Rev. Lett.* **120**, 056801 (2018); Mogi et al. *Sci. Adv.* **3**, eaao1669 (2017)] as the two critical magnetic fields for the zero σ_{xy} plateau are exactly the coercive fields of the Cr or V doped TI layers in the very thick sample limit. When reducing the sample thickness, the width of zero σ_{xy} plateau is also reduced, which can naturally be understood from the coupling between two surfaces. Therefore, the width of the zero σ_{xy} plateau measured at $V_g = V_g^0$ can be used to estimate the coupling strength between the top and bottom surface magnetic TI layers. In our experiments, the width of the zero σ_{xy} plateau becomes broader as m increases and reaches saturation for $m \geq 12$ (Figs. 3a to 3c of the main text).

Fig. R2| Gate dependence of the nonlocal resistance in the $m=100$ sample with antiparallel and parallel magnetization alignments. a, reused from Fig. R1a. **b, c**, V_g dependence of $\rho_{16,34}$ under antiparallel (b) and parallel (c) magnetization alignments. All measurements are taken at $T = 70$ mK and $\mu_0 H = 0$ T.

We did perform nonlocal transport measurements on the $m = 100$ sample (Fig. R2). For the antiparallel magnetization alignment, the value of $\rho_{16,34}$ reaches its maximum at $V_g = V_g^0$ (i.e., the

axion insulator phase). At first glance, this observation suggests the emergence of chiral edge currents in the axion insulator phase, as discussed in Li et al. *Sci. Bull.* **68**, 1252 (2023). However, we would like to point out that the inherent “pickup” resulting from the extremely high ρ_{xx} value in the axion insulator state, together with the imperfect geometry of the Hall bar devices, can easily lead to a higher nonlocal signal. We note that the thick magnetic TI sandwiches used in our study might deviate from the regime discussed in recent theoretical papers [Chen et al. *Phys. Rev. B* **103**, L241409 (2021); Gu et al. *Nat. Commun.* **12**, 3524 (2021); Zou et al. *Phys. Rev. B* **105**, L201106 (2022); Gong et al. *Natl. Sci. Rev.* **10**, nwad025 (2023);]. To explore the existence of the half-quantized counter-propagating Hall current in axion insulators, one should examine a thick sample characterized by dimensions in the mesoscopic range, as opposed to the macroscopic range. Furthermore, the existence of the zero Hall resistance plateau in MnBi_2Te_4 thin flakes as a signature of the axion insulator state remains a subject of ongoing debate [Liu et al. *Nat. Mater.* **19**, 522 (2020); Lin et al. *Nat. Commun.* **13**, 7714 (2022); Chang, Liu, and MacDonald, *Rev. Mod. Phys.* **95**, 011002 (2023)]. We added Fig. R2 and relevant discussion in the revised Supplementary Information.

Comment 3:

(3) The author uses Hall/longitudinal conductance/resistance to characterize the system. However, in my opinion, the conductance/resistance varies when changing the system size. It may be better to use the conductivity/resistivity to characterize the system. This aspect may be much more complex for a 3D system, with the chiral currents located on either the 1D hinge or the 2D side surface.

Response: For the QAH and axion insulator states in thick magnetic TI sandwiches, the conduction is primarily edge/surface-dominated, while the interior bulk remains insulating. Therefore, it is scientifically challenging to accurately calculate the resistivity/conductivity. Therefore, we employed the resistance/conductance rather than the resistivity/conductivity to characterize the conduction characteristics of the samples. As noted in the Methods section, all Hall bar devices used in our transport measurements share the same aspect ratio, with an effective area of $\sim 1 \text{ mm} \times 0.5 \text{ mm}$.

Comment 4:

(4) The author said, “We also find that the two terminal resistance in the axion insulator regime

decreases nearly exponentially with increasing m , implying the rapid reduction of the transport gap, which presumably originates from the side surface states in thick samples“. Why the “nearly exponentially” decay implies a reduction of the transport gap?

Response: The connection between the nearly exponential decay of the two-terminal resistance and the reduction of the transport gap can be explained as follows: In the axion insulator regime of thick samples, the longitudinal transport is mainly determined by the side surface states, which represent the lowest energy states in the system. The side surfaces are gapped due to the finite size effect. For thinner samples, the side surface gap is large. On the other hand, in the 3D limit, the side surface states become metallic without a gap. Therefore, as m increases, there is a crossover from a fully gapped state to a metallic state. Figure 5e of the main text shows the side surface state gap as a function of m . Besides the numerical results, we also analytically show that the side surface gap is proportional to m^{-1} for large m . The analytical results agree well with the numerical results.

Due to the side surface gap, the longitudinal resistance can be approximated by a formula for a thermally-activated semiconductor, which is proportional to $\exp(\delta/k_B T)$. Here δ represents the minimal gap in the side surface states and can be considered as the transport gap. As a result, the longitudinal resistance should decrease with increasing m as $\exp(K/k_B T m)$, with K being a constant. From this point of view, in the logarithmic scale, the resistance is also proportional to m^{-1} , which matches qualitatively with the experimental data well (Fig. 5e of the main text). Therefore, we suggest that the nearly exponential decay of the longitudinal resistance with increasing m comes from the reduction of the side surface gap, which is inversely proportional to m .

We apologize to Reviewer #1 for the confusion induced by our use of the term “*nearly exponential decay*”, as the two-terminal resistance follows an exponential decay as $\exp(K/k_B T m)$, rather than the more common $\exp(-K m)$. We rewrote this sentence in the revised manuscript and added relevant discussion in Supplementary Information.

Comment 5:

(5) I suppose that the side surface gap should exhibit a power-law decay with the increase in the film thickness, which can be analytically confirmed by the quantum well approximation.

Response: Reviewer #1 is correct and the side surface gap δ exhibits a proportional relationship with $\frac{1}{d}$. This power-law decay has a power of 1 as the thickness approaches a large limit. We have

established, both through numerical simulations and analytical approaches, that δ is proportional to $\frac{1}{m}$ (Fig.5e of the main text).

By employing a model similar to that used in Zou et al. *Phys. Rev. B* **105**, L201106 (2022), which accounts for the quantum well confinement of isotropic Dirac fermions, we can derive an equation for the side surface gap as follows: $\delta \tan\left(\frac{\delta d}{2\hbar v_f}\right) = \sqrt{4M^2 - \delta^2}$, where δ is the minimal band gap in the side surface states, v_f is the chemical potential of the Dirac surface states, d is the thickness, and $2M$ is the magnetic exchange gap at the top and bottom surfaces.

For the limit of large M , δ saturates at $\delta_{\max} = \frac{\pi\hbar v_f}{d}$, as anticipated by Reviewer #1. In our manuscript, we used a more realistic tight-binding model that takes into account the anisotropy of the side surface states, to numerically calculate the side surface gap. We note that the analytical results from the simplified model align qualitatively with the numerical simulations. We cited Zou et al. *Phys. Rev. B* **105**, L201106 (2022) in the revised manuscript.

Reviewer #2

This manuscript studies axion insulator state in magnetically doped topological insulator (TI) sandwiches of V-doped (Bi,Sb)2Te3/(Bi,Sb)2Te3/Cr-doped (Bi,Sb)2Te3 with varying the middle undoped TI layer thickness up to 100 nm. The zero Hall conductance plateau is now regarded as insufficient evidence of the axion insulator state because identical features can be seen by other reasons such as the existence of the hybridization gap in the 2D regime. The observation, in this manuscript, of the zero Hall conductance plateaus from the thick samples in the 3D regime provide further evidence of the axion insulator states in the magnetically doped TI sandwiches. The results would be useful to the community. However, I would suggest publication of this manuscript in the Nature Communications after the authors address my comments and questions:

We thank Reviewer #2 for his/her concise summary and appreciation of our work.

Comment 1:

One of the main concerns in the 3D regime is conduction through the bulk and side surfaces, which need to be insulating to observe well-quantized quantum anomalous hall (QAH) “insulator” and

axion “insulator” states. The authors stated that all the observed data are at the charge neutral point by tuning the back gate voltage. Also, they claim that the side surface states are gapped and insulating, and the energy gap opening for the side surface states that they discuss in Fig. 5 is less than 10 meV (1.24 meV for 100nm sample). Since the gap is small, this may need to be better supported by experimental evidence.

Fig. R3| Estimate of the effective activation gap E_a of the $m=100$ sample. $\sigma_{xx}(0)$ as a function of $1/T$. The dashed lines show the fit of the Arrhenius function $\sigma_{xx} = \sigma_{xx}^0 e^{-\frac{E_a}{k_B T}}$. The fit temperature range is 0.07~1.7 K.

Response: As shown in Figs. 1b and 1c of the main text, we have measured the $\mu_0 H$ dependence of ρ_{xx} and ρ_{yx} at $V_g = V_g^0 = 0$ V and different temperatures. Through these magneto-transport results, we plotted $\rho_{yx}(0)$ and $\rho_{xx}(0)$ as a function of T at $\mu_0 H = 0$ T (Supplementary Fig. 2). At low temperatures, the transport behavior is primarily determined by the effective activation gap E_a , which can be described using the Arrhenius equation: $\sigma_{xx} = \sigma_{xx}^0 e^{-\frac{E_a}{k_B T}}$. To estimate E_a , we plotted $\sigma_{xx}(0)$ as a function $1/T$ on the logarithmic scale and performed linear fitting for the $m = 100$ sample (Fig. R3). The estimated value of E_a is $\sim 33.6 \mu\text{eV}$ for the $m = 100$ sample. This further supports our theoretical analysis that the gap of the side surfaces in our thick magnetic TI sandwiches is small. The experimental value of E_a is smaller than the theoretically estimated surface gap δ because the disorder scattering at the side surface can reduce the transport gap. The value of V_g^0 is

determined when the two-terminal resistance $\rho_{16,16}$ is maximized for the antiparallel magnetization alignment (Fig. R1b), see more details in our response to **Comment 1** of Reviewer #1 above. We added Fig. R3 and relevant discussion in the revised Supplementary Information.

Comment 2:

1. 100nm (Bi,Sb)₂Te₃ is not quite thin for back gating. Gating near the bottom layers is more efficient than gating near the top layers. At the charge neutral point, would the chemical potential near the bottom layers and near the top layers be the same for thick samples? Or more p-type near the bottom layers, and more n-type near the top layers, for example? Showing gate-dependent ρ_{xx} and ρ_{yx} will be helpful for the authors' statement regarding the bulk conduction.

Response: Reviewer #2 is correct that “*Gating near the bottom layers is more efficient than gating near the top layers.*”. In our experiments, because of the large dielectric constant ϵ of (Bi,Sb)₂Te₃ [$\epsilon \sim 100$ in Yu et al. *Nanotechnology* **24** 015705 (2013)] and its insulating property, the chemical potential difference between the top and bottom surface layers of the $m = 100$ sample under a single bottom gate voltage might be small. We have no direct evidence to show different gating efficiency in magnetic TI sandwiches with different m . We did not measure the gate-dependent ρ_{xx} and ρ_{yx} but measured the gate-dependent of the two-terminal resistance $\rho_{16,16}$ of the $m = 100$ sample with antiparallel and parallel magnetization alignments (Fig. R1). We added Fig. R1 and relevant discussion in the revised Supplementary Information.

Comment 3:

2. It is stated that the energy gap for the side surface states is smaller in thicker samples ($\sim 1\text{meV}$ for $m=100$ sample and $\sim 6\text{meV}$ for $m=8$ sample). And, well-quantized QAH and axion insulator states are expected when the chemical potential is located within the gap. Narrower gate-voltage range for well-defined quantization due to smaller gap is anticipated in thicker samples. Can this be provided?

Response: As noted in our manuscript, the magnetic TI sandwich samples used in this work are grown on heat-treated insulating SrTiO₃ (111) substrates. The thickness of the SrTiO₃ substrate is ~ 0.5 mm. Because of its huge dielectric constant at low temperatures ($\epsilon \sim 18000$ at liquid helium temperature) [Sakudo and Unoki, *Phys. Rev. Lett.* **26**, 851 (1971)], the ~ 0.5 mm SrTiO₃ (111) substrate can be used as the dielectric to apply a bottom gate to tune the chemical potential of the magnetic TI sample [Chang et al. *Adv. Mater.* **25**, 1065 (2013); Chang et al. *Science* **340**, 167

(2013); Chang, Liu, and MacDonald, *Rev. Mod. Phys.* **95**, 011002 (2023)]. However, the dielectric constant ϵ of the SrTiO₃ substrate exhibits variations at different V_g [Caviglia et al. *Nature* **456**, 624 (2008)]. Therefore, since magnetic TI sandwiches with different m usually have different V_g^0 , the width of the quantized plateau does not represent the transport gap size (i.e., the gap of the side surface states). In our experiments, we do not observe a systematic decrease in the plateau width as m increases.

Comment 4:

Below are other comments and questions:

3. In page 8 line 156, it is stated that “the side surface states are insulating”. The wording of “surface states” and “insulating” sounds contradictory.

Response: Corrected.

Comment 5:

4. In page 8 line 167-169, “Therefore, the appearance of the zero σ_{xy} plateau in our priors studies should be a result of the antiparallel magnetization alignment rather than the formation of the hybridization gap in the $m=4$, $m=5$, and $m=6$ samples.” - Do all the samples in the previous studies have the sample details and growth conditions such as doping concentrations, gate dependence, and so on? Otherwise, results in this manuscript cannot be generalized for all the previous studies by the authors.

Response: Yes, we employed the same growth recipes [e.g., the growth temperature, the element concentrations, the growth rate, etc.] for the samples used in this work and our prior two works [Xiao et al. *Phys. Rev. Lett.* **120**, 056801 (2018); Wu et al. *Nat. Commun.* **11**, 4532 (2020)]. We added this information in the Methods section of the revised manuscript.

Comment 6:

5. From the two-terminal resistance measurements, it is stated that $\rho_{12,12}$ in the QAH regime is $\sim h/e^2$ for all samples. What is the contact resistance? Were there any additional series resistance, for example from the electronic filters in the cryostats? Did the authors subtract any series resistance?

Response: As noted in the Methods section of our manuscript, we used a dilution refrigerator (Oxford Instruments, 70 mK, 8 T) to perform the two-terminal measurements. The contact

resistance including wires used in our dilution fridge at room temperature is $\sim 34 \Omega$. No electronic filter is involved in this dilution refrigerator. We didn't subtract any series resistance.

Comment 7:

In page 9 line 176, “near-plateau feature” – It looks flat in logarithmic scale, but it does not look like a plateau in linear scale.

Response: Reviewer #2 is correct that $\mu_0 H$ dependence of $\rho_{12,12}$ for the $m = 75$ and $m = 100$ samples does not look like a plateau in linear scale. The data of the $m \geq 8$ samples in linear scale have been shown in Supplementary Fig. 10. As m increases, the samples show a systematic evolution from a peak feature to a nearly flat feature. For the benefit of readers who may share the same concerns as Reviewer #2, we rewrote this sentence in the revised manuscript.

Comment 8 :

7. In page 11 line 224, “the confinement-induced gap should be the main origin of the side surface gap” – This statement is too strong since it just came from the qualitative similarity of the decaying delta in linear scale with the decaying resistivity in logarithmic scale.

Response: As noted in our response to **Comment 4** of Reviewer #1, the connection between the reduction of the transport gap in linear scale and the nearly exponential decay of the two terminal resistances in logarithmic scale can be explained as follows: In the axion insulator regime, the longitudinal transport is mainly determined by the side surface states, which represent the lowest energy states in the system. However, since the side surface states still exhibit a gap, we can approximate the longitudinal resistance using the formula for a thermal-activated semiconductor, which is proportional to $\exp(\delta/k_B T)$. Here δ represents the minimal gap in the side surface states and is considered as the transport gap.

Our theoretical analysis reveals that the confinement-induced gap in the side surface states is approximately proportional to m^{-1} for large m . If we assume that the transport gap δ arises from the confinement effect, then longitudinal resistance should decrease with increasing m as $\exp(K/k_B T m)$, where K is a constant. When plotted on a logarithmic scale, the resistance also exhibits a proportional relationship with m^{-1} , which qualitatively aligns with the experimental data (Fig. 5e of the main text). Therefore, if the transport gap δ is indeed a consequence of the confinement effect, the nearly exponential decay of the two terminal resistances with increasing m may be attributed to the reduction of the transport gap, which inversely scales with m .

Moreover, one more piece of evidence also indicates that the transport gap may arise from the confinement effect, as demonstrated by the plateau feature of $\rho_{12,12}$ in Fig. 4a of the main text for the $m = 75$ and $m = 100$ samples. As noted in the manuscript, this behavior can be attributed to the dependence of δ on $2M$ (Fig. 5d of the main text). Note that $2M$ is directly linked to the external magnetic field, whereas δ is not. For the $m = 75$ and 100 samples, which are located within the saturated region and δ is insensitive to variation in $2M$, they exhibit a nearly flat behavior against the external magnetic field on the logarithmic scale. In contrast, for thinner samples, $\rho_{12,12}$ shows a sharp peak (Supplementary Fig. 10). This evidence also supports the claim that the confinement-induced gap is the primary origin of the side surface transport gap.

We rewrote this sentence to convey a less assertive statement in the revised manuscript and added the relevant discussion in Supplementary Information..

Comment 9:

8. EDS mapping of V (green) and Cr (red) near V-doped and Cr-doped regions is shown in Fig. 1a. The colors in the doped layers and supposedly undoped layers look similar. Does this mean that the dopants diffused into the undoped layers?

Fig. R4| Cross-sectional STEM and the EDS map of the $m=100$ sample. a, b, Cross-sectional STEM image (a) and the EDS map of V (b) near the top surface layers of the sample.

Response: The quality of the EDS map is influenced by both the signal strength and the cleanliness of the spectrum. For our 3 QL $(\text{Bi,Sb})_{1.89}\text{V}_{0.11}\text{Te}_3/m$ QL $(\text{Bi,Sb})_2\text{Te}_3/3$ QL $(\text{Bi,Sb})_{1.85}\text{Cr}_{0.15}\text{Te}_3$ sandwiches, the doping level of V in the top 3 QL layer is ~5%, this diluted doping makes it challenging to distinguish the V signal from the background signals. One prior study has also shown similar EDS map results [Mogi et al. *Sci. Adv.* **3**, eaao1669 (2017)]. Nevertheless, by narrowing down the selected energy window, we can further minimize the impact of the background noise (Fig. R4). We used the new EDS spectra of V to replace the old ones in Fig. 1 in the revised manuscript.

Comment 10:

9. In the supplementary Figs. 4-7 for $m=3$ to 30, some of the ρ_{yx} curves look very noisy and they go out of the given y-axis. How much do they deviate? What is the interpretation of the huge noise in the samples while some are not as noisy? Can this be replotted and seen in the full y-axis range instead of cropping them within $\pm 1 h/e^2$?

Response: We apologize to Reviewer #2 for the confusion and thank him/her for bringing up this issue. In the axion insulator regime, the value of ρ_{xx} is extremely huge, which is much greater than the value of ρ_{yx} . For the Hall bar devices used in our transport measurements, the contacts for measuring ρ_{yx} might not be well aligned. The small pickup between ρ_{xx} and ρ_{yx} can make ρ_{yx} values noisy, which is not easy to eliminate by symmetrizing the data under positive and negative $\mu_0 H$ [Xiao et al. *Phys. Rev. Lett.* **120**, 056801 (2018); Wu et al. *Nat. Commun.* **11**, 4532 (2020)]. Moreover, a prior study has pointed out the finite capacitive coupling between the electrodes can also cause interference between ρ_{xx} and ρ_{yx} , resulting in noisy ρ_{yx} in the axion insulator regime [Mogi et al. *Sci. Adv.* **3**, eaao1669 (2017)].

We plotted the full range of $\rho_{yx}-\mu_0 H$ curves and added this figure in the revised Supplementary Information.

Reviewer #3

This manuscript presents the demonstration of an axion insulator state in a 3D topological insulator heterostructure and the investigation of its properties as a function of thickness. This investigation is made possible by the MBE growth of a heterostructure with an insulating layer of variable thickness separating top and bottom layers doped with two different magnetic elements,

permitting the realization of antiparallel magnetization orientations. The existence of the axion insulator state is verified by the presence of the quantum anomalous Hall effect along with a zero Hall conductance plateau near field. Varying the intermediate insulating layer thickness shows that the conditions for the axion insulator state disappear in the 2D limit of 1 QL. The authors also claim that the presence of the axion insulator state is dependent upon there being a gap in the intermediate layer side surface states.

We thank Reviewer #3 for his/her concise summary of our work and thoughtful comments.

Comment 1:

The novelty of this work lies in the demonstration of the axion insulator state in thick 3D materials well away from the 2D limit (i.e. $m \gg 10$). Overall, the study thoroughly investigates the effect of the thickness of the intermediate layer on the transport behavior, and the manuscript presents a clear discussion of the origin of the phenomena that are experimentally observed. Furthermore, theory is used to support the assertion that the intermediate layer side surface states are gapped. While the manuscript points out that this study is unique in that it investigates the 3D axion insulator state in samples with thicknesses greater than 10 nm, The study itself indicates that this state is present in samples with thicknesses greater than 3 nm. To strengthen the novelty of this study, a discussion on the new information that is gained over those previous studies should explicitly be included.

Response: We thank Reviewer #3 for his/her positive assessment of our work. We followed Reviewer #3's suggestions and added "*a discussion on the new information that is gained over those previous studies*" in the summary paragraph of the revised manuscript.

Comment 2:

The methodology appears to be sound, although more information should be included about the heterostructure to enable replication. The concentrations of V and Cr doping should be included, and the rationale for selecting 3 QL thicknesses for each of these doped layers should be noted.

Response: We apologize to Reviewer #3 for not including this information in our original manuscript. We added the doping concentrations of V and Cr and the Bi/Sb ratio in the Methods section of the revised manuscript.

In our experiments, to achieve satisfactory magnetization, a minimum thickness of 3 QLs is required for both the bottom Cr-doped and the top V-doped TI layers. The use of thicker Cr/V-

doped TI layers carries the risk of introducing additional disorders, leading to an inevitable degradation in sample quality. After considering these factors, we decided to employ a thickness of 3 QL for the Cr/V-doped TI layers, enabling us to realize both the axion insulator state and the QAH state. We added this information in the Methods section of the revised manuscript.

Comment 3:

A few additional minor questions should be addressed for improved comprehension:

- *The experimental and theoretical investigation included samples with m up to 100. However, the theoretical calculations suggest that the gap in the side surface states (a condition for this effect) starts to close around $m = 100$. At what thickness does the 3D axion insulator state expected to cease to be observed?*

Response: We employed a phenomenological formula to estimate the two-terminal resistance $\rho_{12,12,\max} \sim K_1 \exp(K_2/m)$ for large m . Through fitting the experimental data, we determined the values of K_1 and K_2 are $\sim 3.7h/e^2$ and 330, respectively. When the side surface gap δ is comparable with $k_B T$, the side surface should be metallic rather than insulating, and then the axion insulator state is no longer expected to be observed. The transition is given by the phenomenological formula, where $K_2/m \sim 1$. Therefore, the critical thickness is $m \approx K_2 = 330$. We added this discussion in the revised Supplementary Information.

Comment 4:

- *“ $2M$ ” should be defined in the text.*

Response: Done.

-----List of changes-----

(All the changes in the main article are shown in blue)

1. Line 59 on Page 3, we rewrote the below sentence.

“specifically, 3 quintuple layers (QL) $(\text{Bi,Sb})_{1.89}\text{V}_{0.11}\text{Te}_3/m$ QL $(\text{Bi,Sb})_2\text{Te}_3/3$ QL $(\text{Bi,Sb})_{1.85}\text{Cr}_{0.15}\text{Te}_3$.”

2. Line 65 on Page 3, we rewrote the below sentence.

“We also find that the two-terminal resistance in the axion insulator regime decreases rapidly with increasing m . We perform theoretical calculations on the side surface gap δ and find that its decay behavior with increasing m is consistent with our experimental observation.”

3. Line 176 on Page 9, we rewrote the below sentence.

“However, for the $m = 75$ and $m = 100$ samples, $\rho_{12,12}$ changes slightly and shows a nearly flat feature as a function of $\mu_0 H$ in the axion insulator regime on a logarithmic scale (Fig. 4a).”

4. Line 190 on Page 9, we added the below sentence.

“To examine the QAH and the axion insulator phases in thick magnetic TI sandwiches, we measure the V_g dependence of both the two-terminal resistance and nonlocal resistance in the $m = 100$ sample when its top and bottom magnetic layers exhibit antiparallel and parallel magnetization alignments (Supplementary Figs. 11 and 12).”

5. Line 215 on Page 10, we rewrote the below sentence.

“Figure 5d shows the side surface energy gap δ as a function of the magnetic exchange gap $2M$ for $m \geq 8$.”

6. Line 218 on Page 11, we rewrote the below sentence.

“These two regimes provide an understanding of the nearly flat feature observed in thick axion insulators and the peak behavior in thin axion insulators (Fig. 4a and Supplementary Fig. 13).”

7. Line 235 on Page 11, we rewrote the below sentences.

“The axion electrodynamics from the bulk θ -term, which is unique in 3D, gives rise to many topological responses such as the topological magnetoelectric effect²⁻⁴, image magnetic monopole³⁵, and quantized optical response². Our hundred-nanometer-thick magnetic TI sandwiches with the axion insulator state in the 3D regime (~ 106 nm thick) provide a better material platform for the exploration of these topological responses^{2-4,35}, as well as the higher-

order TI phase^{14,15}. Moreover, our thick magnetic TI sandwiches can also be employed to explore the existence of the half-quantized counter-propagating Hall current in axion insulators³⁶⁻³⁹.”

8. Line 247 on Page 12, we added the below sentence.

“To achieve satisfactory magnetization, both the top and bottom 3 QL (Bi,Sb)₂Te₃ layers doped with Cr or V are employed.”

9. Line 270 on Page 13, we added the below sentence.

“The contact resistance including wires used in our dilution fridge at room temperature is ~34 Ω. No electronic filter is involved in our dilution refrigerator.”

10. We added the below sentence in the caption of Fig.4.

“No series resistance is subtracted in these two-terminal measurements.”

11. We replotted Fig. 1a in the revised manuscript.

12. We added Supplementary Figs. 8, 11, and 12 in Supplementary Information.

13. We added Sections II.1 to II.3 in Supplementary Information.

14. We added numbers of references shown in blue in the revised manuscript shown in blue and Supplementary Information.

15. We made numbers of minor revisions shown in blue in the revised manuscript and Supplementary Information.

REVIEWERS' COMMENTS

Reviewer #1 (Remarks to the Author):

There are still many issues in the field of axion insulators that require further research. The authors have identified a zero plateau in a relatively thick sample, providing evidence for the existence of axion insulators. The research has taken a significant step in the research on axion insulators.

The authors have successfully replied my comments and made some refinements to the presentation of the results in the revised version of the manuscript, thus improves the quality of the manuscript. Now, I recommend the paper for publication in Nature Communications.

Reviewer #2 (Remarks to the Author):

The authors addressed my comments and questions to my satisfaction. I recommend publication of this manuscript.

Reviewer #3 (Remarks to the Author):

The authors have sufficiently addressed my comments.